# Research on the mechanism of multilayer spiral fog screen dust removal at the comprehensive excavation face

**Deji Jing**[1,2,3]*, **Zhen Li**[1,2,3], **Shaocheng Ge**[4], **Tian Zhang**[1,2,3], **Xiangxi Meng**[1,2,3], **Xin Jia**[1,2,3]

**1** College of Safety Science and Engineering, Liaoning Technical University, Fuxin, China, **2** Research Institute of Safety Science and Engineering, Liaoning Technical University, Fuxin, China, **3** Thermodynamic Disasters and Control of the Ministry of Education, Liaoning Technical University, Fuxin, China, **4** Safety and Emergency Management Engineering College, Taiyuan University of Technology, Taiyuan, China

* jingdeji@163.com

**Data Availability Statement:** All relevant data are within the paper.

**Funding:** National Natural Science Foundation of China Youth Fund Grant (51704146); Liaoning

## Abstract

To solve the problem of the inability of traditional spray dust removal technology to efficiently restrain dust diffusion at the heading face, a multilayer spiral fog curtain dust control method based on spirally arranged pneumatic nozzles is proposed. In this paper, the k-ε turbulence model and K-H droplet breakage model are used. First, different airflow fields are analyzed by simulating the simultaneous injection of different numbers of nozzles, and the motion law of airflow interaction is obtained. Taking the two-layer fog curtain as an example, a multiphysical field coupling numerical simulation of the two-layer spiral fog curtain applied in the field is carried out, and the variation law of its velocity field distribution and particle motion characteristics is analyzed. A similar experimental platform is established to verify the effectiveness of the simulation results and the feasibility of the dust removal scheme. The simulation results show that the double helix arrangement will form a rotating airflow with the cutting arm as the axis to cover the whole roadway section and produce a double-layer spiral fog curtain. The water mist is fragmented into smaller fog droplets under the action of rotating airflow, which improves the dust catching effect of the fog curtain. Experiments show that the dust removal rate and efficiency of multilayer spiral fog curtains are obviously stronger than those of natural dust reduction and traditional spray. After 3 minutes, a dust concentration of approximately 470 mg/m$^3$ can be reduced to less than 4 mg/m$^3$. The average dust removal rates of total dust and exhaled dust were 2.600 mg/(m$^3$.s) and 0.189 mg/(m$^3$.s), respectively, and the dust removal efficiencies were 97.01% and 94.32%.

## 1. Introduction

The tunneling face is one of the main dust-producing areas in mines. With the improvement of the mechanization and automation of underground mining operations, the amount of dust produced during tunneling construction has increased sharply [1–3]. According to field measurements, the dust concentration of key dust-producing sources in a fully mechanized

Provincial Natural Science Foundation Project (2020-MS-304); Liaoning Provincial Education Department Scientific Research Funding Project (LJKZ0323).The funders had no role in study design, data collection and analysis, decision to publish, or preparation of the manuscript.

tunneling face can reach 2000–2500 mg/m$^3$ without dust prevention measures. These high-concentration dusts increase the probability that field workers will suffer from pneumoconiosis, which seriously affects the safe production of mines and the physical and mental health of workers [4–7].To realize low mine dust production, scholars both domestically and abroad have performed mechanistic research and field practice studies [8–11]. Xiao et al. [12] researched the coupled migration law of airflow and dust in the limited space of a large section roadway and used Fluent to simulate the discrete phase model (DPM) under normal pressure and short ventilation, determined the optimal dust removal parameters, and verified the simulation reliability through field research. Jiang et al. [13] experimentally studied the atomization parameters of gas-water nozzles, established the mathematical model of the dust reduction process of gas-water spray, and obtained the dust reduction efficiency curve of gas-water nozzles, which provided a reference for selecting the best gas-water flow rate. Huang et al. [14] designed a graded and zoned high-efficiency dust removal system to use different dust removal methods to control dust in three areas before, during and after the integrated excavation work respectively, and finally, the dust removal performance of the equipment was measured on site. Cheng et al. [15] experimentally measured the dust characteristics of different mine production sites and the atomization characteristics of different types of nozzles at different pressure conditions, and conducted comparative research in field applications. It was concluded that the optimization of spray parameters and the selection of nozzle types can effectively improve the dust removal efficiency and the corresponding optimal spraying parameters at different positions. Nie et al. [16] developed a new type of external spray dust removal device, obtained the optimal spray parameters of the device through numerical simulation analysis and experimental research, and conduct field tests on the dust removal performance of the new device. In the existing technical research, most dust removal devices have single spray or air curtain designs, and research combining a spiral air curtain and spray dust removal is still very limited.

Therefore, to further improve the dust removal ability of the heading face, a multilayer vortex mist curtain dust removal method is proposed and combined with a cyclone fog curtain dust removal scheme. Multinozzles are arranged in a multilayer and outward diffusion spiral arrangement, and gas-water two-phase spray mode flow is adopted. The running track and mutual influence of air flow and water mist emitted from the nozzles are studied using this mode. The results are analyzed through multiphysical field coupling simulation calculations, which provide a basis for the testing and application of this dust removal scheme. Finally, by building a similar experimental platform, the dust removal effects of natural sedimentation, a traditional spray dust removal scheme and the multilayer vortex mist curtain dust removal scheme are compared to provide theoretical guidance for the field engineering application of this dust removal scheme.

## 2. The dust removal method using a multilayer spiral fog curtain is proposed

With the existing dust removal technology conditions, the dust concentration and coal dust properties of the heading face of Mindong No. 1 Mine of Inner Mongolia Mengdong Energy Co., Ltd. were tested on site from October to December 2019. Under the current production intensity and operation(and failure) conditions of dust removal facilities, the test results mainly showed tunneling in coal roadways and a small part of rock roadways. The particle size distribution of rock fractures consisted mainly of large particles. In addition, the water content was high, the heading face was well ventilated, and the dust concentration in the nonworking state was extremely low. However, the dust pollution at the head of the heading machine was

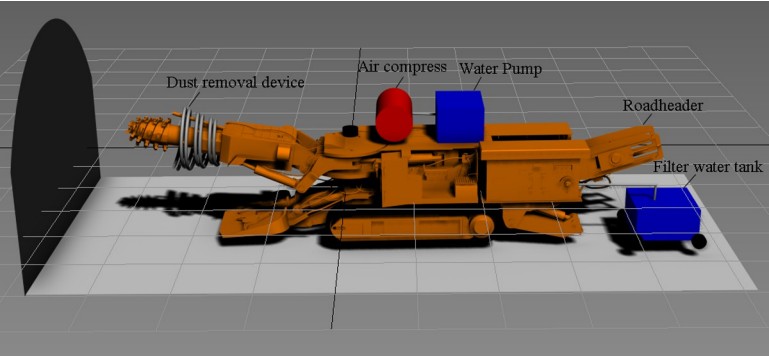

**Fig 1. 3D design drawing of the multilayer spiral fog curtain dust removal device.**

extremely serious during the heading process, and the dust concentration was as high as 2000–2500 mg/m³, which cannot be ignored, and the impact on the health of underground workers cannot be ignored. The comprehensive treatment of dust at the main dust source points in the heading face is an urgent problem to be solved. In view of this, a new dust removal method for multilayer vortex mist curtains is proposed and provides new ideas for dust control at heading faces.

## 2.1 Composition of the device

The practical design of the device is shown in Fig 1. A pipeline bracket for fixing pipelines and nozzles is installed on the cutting arm of the roadheader. One end of the air supply pipe is connected to an air compressor, the other end of the air supply pipe is spirally fixed on the pipeline bracket, one end of the water supply pipe is connected to a water pump, the other end of the water supply pipe is spirally fixed on the pipeline bracket, and both the air compressor and the water pump are installed on the roadheader body. The spiral water supply pipe and the air supply pipe are connected with a plurality of water-gas intersection pipes, and the nozzle is installed on the water-gas intersection pipe. The spray direction of the nozzle can be adjusted by rotating the water-gas intersection pipe. The assembly structure of the nozzle is shown in Fig 2.

The multilayer spiral fog screen dust removal device provides spray power through an air compressor and water pump, and different spray effects can be realized by setting parameters, such as pressure and flow rate. The pipeline bracket can also be adjusted radially and axially

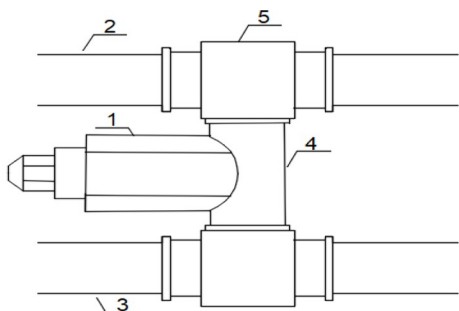

**1-Nozzle 2-Water Supply Pipe 3-Air Supply Pipe 4-Water-Gas Interchange Pipe 5-Adjustable Connector**

**Fig 2. Schematic diagram of the nozzle assembly.**

according to the field working environment to achieve the best fog screen state. The fixed position and number of nozzles on the bracket can also be changed to adjust of the distance between the fog curtains and the range of the water mist. In the device construction scheme, which can form a multilayer fog curtain, the method of arranging nozzles in a spiral manner is simple in structure and easy to realize. In addition, the structure can make the sprayed water mist have a forward partial speed, which is beneficial to the forward advancement of the water mist. The effective range of dust removal can be changed with the position of the roadheader head, and the dust removal capacity will not be affected by changes in the dust production at different points, thus further improving the comprehensive dust removal effect of the heading face.

## 2.2 Working principle

After the installation of the device on the roadheader, before the tunneling task, the water pump and air compressor are turned on. After the spray of the device stabilizes, the bracket structure and the direction of the nozzle can be adjusted according to the field environment to achieve the best working state for forming a multilayer spiral fog curtain. Under the action of the multilayer spiral fog curtain dust removal device, a multilayer spiral fog curtain will be formed around the heading face. The large amount of dust generated during the cutting operation of the roadheader is controlled in a limited space at the head of the roadheader, and the multilayer fog curtain has more barriers to blocking dust than a single-layer fog curtain. Suspended dust is blocked layer by layer. Under the action of the high-efficiency dust-catching ability of broken droplets in airflow, the main dust source on the heading face is effectively isolated from the driver area of the roadheader, which inhibits the diffusion of dust produced on the heading face to the working area and reduces the damage to workers and mechanical equipment during heading. Fig 3 is a 3D schematic diagram of the spraying effect of the multilayer spiral fog curtain device during normal roadheader operation.

## 3. Physical model establishment

Many researchers use comsol software for scientific research, and the simulation software is better able to perform numerical simulations with coupled multi-physical fields and the simulation results are highly reliable [17–19]. Comsol was used to simulate and calculate the airflow

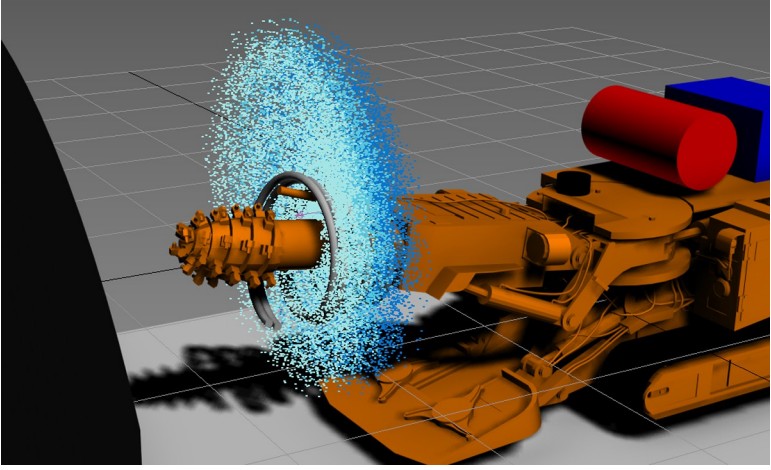

**Fig 3. 3D schematic diagram of the spray effect of the multilayer spiral fog curtain device.**

field, wind speed and particle trajectory of the multilayer spiral spray to explain the influence of this method on the airflow and water mist flow direction and to provide a guaranteed and reliable basis for subsequent test.

## 3.1 Turbulence model

The standard $k$–$\varepsilon$ turbulence model was adopted, as it has high accuracy in calculating the reverse pressure gradient flow field and can be applied to a wall-bound flow; that is, it can be applied to the calculation of the airflow direction and velocity in the tunneling roadway [20–23].

$$\rho(u \cdot \nabla)u = \nabla \cdot [-pl + (\mu + \mu_T)(\nabla u + (\nabla u)^T)] + F$$

$$\rho\nabla \cdot (u) = 0 \tag{1}$$

Assuming that the air in the roadway is incompressible, the transport equation of turbulent kinetic energy $k$ is as follows:

$$\rho(u \cdot \nabla)k = \nabla \cdot \left[\left(\mu + \frac{\mu_T}{\sigma_K}\right)\nabla k\right] + P_k - \rho\varepsilon \tag{2}$$

The transport equation of the turbulent dissipation rate $\varepsilon$ is

$$\rho(u \cdot \nabla)\varepsilon = \nabla \cdot \left[\left(\mu + \frac{\mu_T}{\sigma_e}\right)\nabla\varepsilon\right] + c_{e1}\frac{\varepsilon}{k}P_k - c_{e2}\rho\frac{\varepsilon^2}{k}$$

$$\varepsilon = ep \tag{3}$$

$$\mu_T = \rho c_\mu \frac{k^2}{\varepsilon} \tag{4}$$

$$P_k = \mu_T[\nabla u : (\nabla u + (\nabla u)^T)] \tag{5}$$

where $\rho$ is the density, kg/m$^3$; $t$ is the time, s; $\mu_T$ is the turbulent viscosity coefficient; $u$ is the wind speed, m/s; $P_k$ is the increment of turbulent kinetic energy caused by the mean velocity gradient,J;and $\sigma_k$ and $\sigma_\varepsilon$ are the Planck coefficients for the turbulent kinetic energy $k$ and dissipation rate $\varepsilon$, respectively; $pl$ is the reynolds stress, N; $F$ is the other external forces, N.

## 3.2 Drag model

The drag force model can be used to calculate the movement process of droplets with airflow.
The fluid drag force obeys the Schiller–Naumann law [24–26]:

$$F_D = \frac{1}{\tau_p}m_p(u - v) \tag{6}$$

$$\tau_p = \frac{4\rho_p d_p^2}{3\mu C_D Re_r} \tag{7}$$

$$C_D = \frac{24}{Re_r}\left(1 + 0.15\ Re_r^{0.687}\right) \tag{8}$$

$$Re_r = \frac{\rho |u - v| d_p}{\mu} \tag{9}$$

$F_D$ is the traction force, N; $\tau_p$ is the shear stress, N; $C_D$ is the drag coefficient; $Re_r$ is the reynolds number; $m_p$ is the droplet mass, kg; $u$ and $v$ are the wind speed and droplet motion velocity, m/s, respectively; $\rho_p$ is the droplet density, kg/m3; $d_p$ is the droplet diameter, μm; and $\mu$ is the gas dynamic viscosity, Pa·s.

## 3.3 Droplet breakage model

The K-H fragmentation model is used to explain the instability caused by two moving fluids. When two fluids move parallel and reach a certain relative velocity, they will transition to an unstable state and decompose into smaller droplets [27–29]. Based on this principle, the breakup of water mist can be simulated, and the droplet size can be calculated.

Based on the linear stability analysis of the liquid jet, the radius change rate when the droplet breaks is

$$\frac{\partial r_{KH}}{\partial t} = \frac{r_{ch} - r_{KH}}{\tau_{KH}} \tag{10}$$

where $r_{ch}$ is the radius of the droplet after fragmentation, μm; $r_{KH}$ is the radius of the droplet before fragmentation, μm; and $\tau_{KH}$ is the fragmentation time, s.

The wavelengths of unstable waves $\Lambda_{KH}$ are:

$$\Lambda_{KH} = \frac{9.02 r_p (1 + 0.45\sqrt{Z})(1 + 0.4T^{0.7})}{(1 + 0.865 We_g^{1.67})^{0.6}} \tag{11}$$

Using the discrete equation of the maximum growth rate, the frequency of unstable waves with the fastest growth rate $\Omega_{KH}$ is as follows:

$$\Omega_{KH} = \frac{0.34 + 0.385 We_g^{1.5}}{(1 + Z)(1 + 1.4T^{0.6})} \sqrt{\frac{\sigma_p}{\rho_p r_p^3}} \tag{12}$$

The fragmentation time $\tau_{KH}$ is

$$\tau_{KH} = \frac{3.788 B_{KH} r_p}{\Omega_{KH} \Lambda_{KH}} \tag{13}$$

Among them

$$Z = \frac{\sqrt{We_l}}{Re_l}, T = Z\sqrt{We_g}$$

$$We_l = \frac{\rho_p U_{rel}^2 r_p}{\sigma_p}, We_g = \frac{\rho U_{rel}^2 r_p}{\sigma_p}$$

$$Re_l = \frac{\rho_p U_{rel} r_p}{\mu_p}, U_{rel} = |v - u| \tag{14}$$

$Z$ is the liquid Ohnesorge number, $T$ is the Taylor number, $We_l$ is the liquid Weber number, $We_g$ is the gas Weber number, $Re_l$ is the liquid Reynolds number, $U_{rel}$ is the gas–liquid velocity difference, m/s, $r_p$ is the initial droplet radius, μm, $\rho$ is the gas density, kg/m, $\sigma_p$ is the liquid surface tension N/m, and $\mu_p$ is the liquid dynamic viscosity, Pa·s. The radius of droplets formed after crushing is obtained:

At that time $B_0\Lambda_{KH} \leq r_p$,

$$r_{ch} = B_0\Lambda_{KH} \tag{15}$$

At that time $B_0\Lambda_{KH} \geq r_p$,

$$r_{ch} = min\left(\left(\frac{3\pi r_p^2 U_{rel}}{2\Omega_{KH}}\right)^{\frac{1}{3}}, \left(\frac{3r_p^2\Lambda_{KH}}{4}\right)^{\frac{1}{3}}\right) \tag{16}$$

$B_{HK}$ and $B_0$ are empirical parameters of the model. The value of $B_{KH}$ is related to the nozzle size design and spray state, and the value is usually 10. $B_0$ is used to control the crushing time.

## 3.4 Parameter settings

The following parameters are set according to the working state when the ambient temperature is 20°C. Environmental and spray parameters are set as shown in Table 1.

# 4. Numerical simulation

## 4.1 Numerical simulation study and results of spray airflow analysis

The simulation model is a two-dimensional space with a width and height of 3 m and 2 m, respectively. The vertical boundary in the positive X-direction is set as the outlet, and the airflow will not return when approaching the outlet. The color legend on the right side of the simulation results represents the airflow velocity(m/s). Without the influence of other external forces and conditions, the simultaneous spraying of a different number of nozzles is simulated, and all the nozzles are of the same model. The nozzle whose spraying direction is parallel to the X axis is nozzle No. 1; then, nozzles No. 2 and No. 3 are added in turn, and three different airflow fields are obtained, as shown in Figs 4–6.

As shown in Fig 4, when there is only nozzle 1, the direction of the air flow is only advanced in the direction of the nozzle because there is only air flow in that direction. It shows that when there is only one nozzle, the airflow will be emitted in a straight line almost along the

**Table 1. Working environment parameter setting.**

| Parameter name | Parameter setting |
|---|---|
| Temperature | 20°C |
| Air pressure | 1 atm |
| Average droplet size | 10 μm |
| Wind speed at nozzle | 30 m/s |
| Air density $\rho$ | 1.205 kg/m$^3$ |
| Water density $\rho_p$ | 1000 kg/m$^3$ |
| Aerodynamic viscosity $\mu$ | $1.79 \times 10^{-5}$ Pa/s |
| Hydrodynamic viscosity $\mu_p$ | 0.001 Pa/s |
| Water surface tension $\sigma_p$ | 0.729 N/m |
| Wall setting | Vanish |
| Exit settings | Disappear, inhibit reflux |

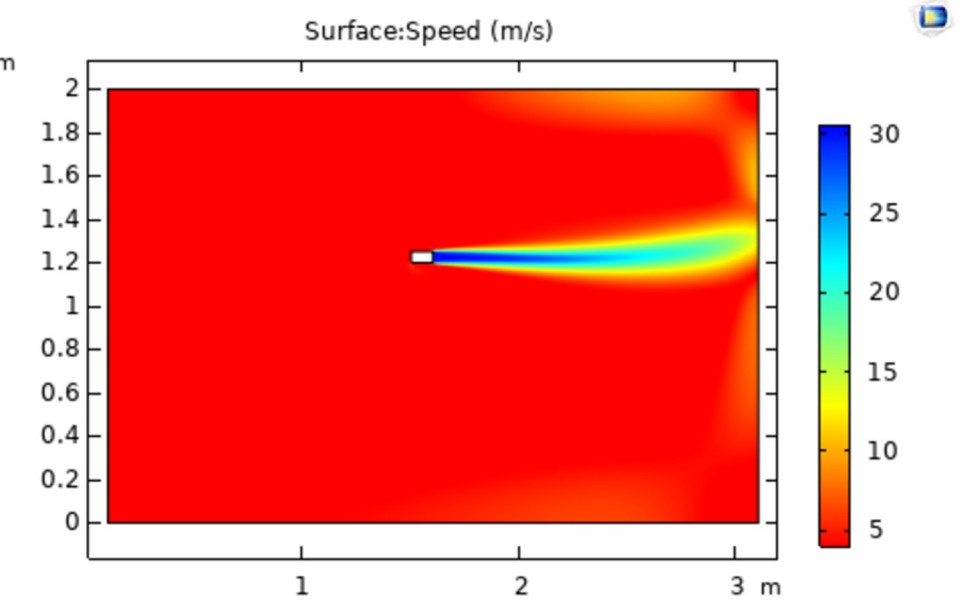

**Fig 4. Flow velocity distribution from nozzle no. 1.**

direction of the nozzle, and the direction will not change. The dispersion degree of the airflow is also small, and the ejected airflow is almost cylindrical.

Next, the air flow field when two nozzles spray at the same time was simulated: nozzle 2 was added in the negative X direction from nozzle 1. The two nozzles are adjacent and tangent in an arc direction, and the angle of the injection direction is 40 degrees different, thus obtaining the air flow field shown in Fig 5. The injection direction of the two nozzles has shifted to a certain extent. After the airflow of nozzle 2 passes through the airflow of nozzle 1, the injection

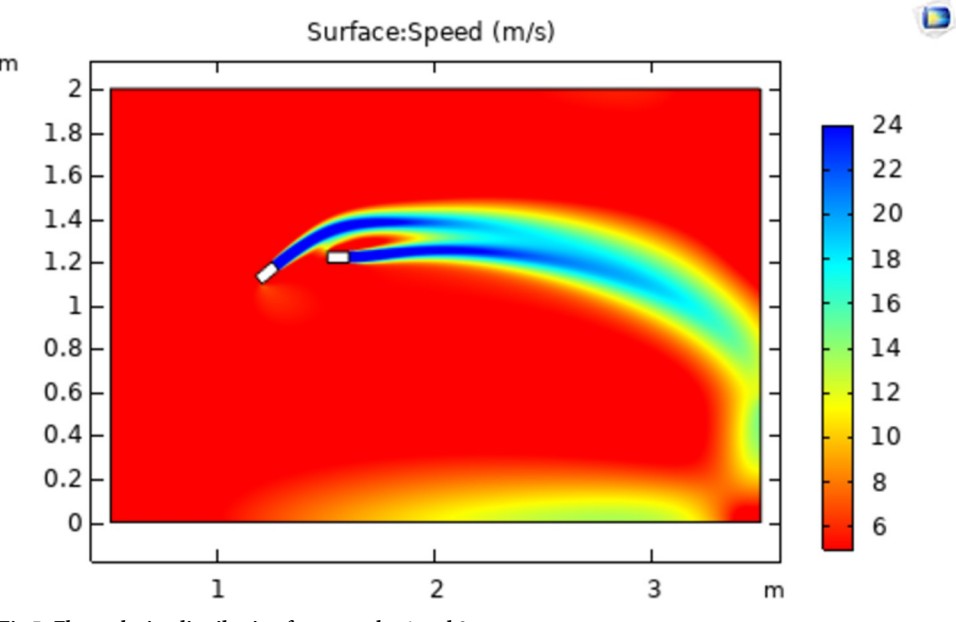

**Fig 5. Flow velocity distribution from nozzles 1 and 2.**

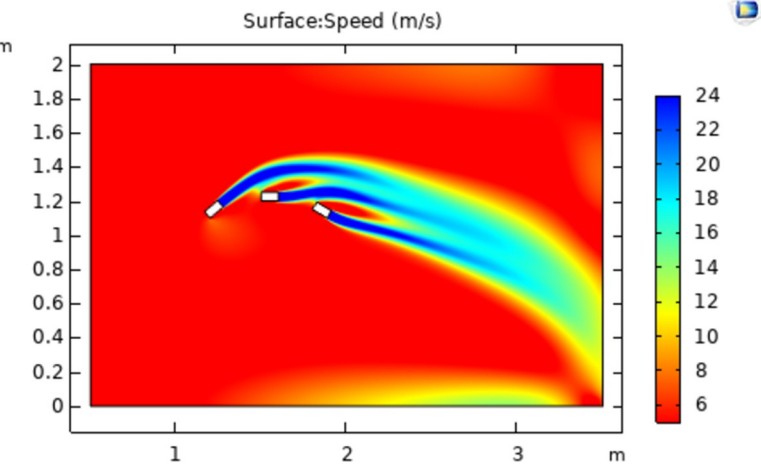

**Fig 6. Flow velocity distribution from nozzles 1–3.**

direction approaches the injection direction of nozzle 1 and shifts downward. The airflow direction of nozzle No. 1 should be parallel to the X axis. Due to the influence of other airflows, the initial injection direction shifts slightly upward, and the airflows in the two directions tend to merge into a whole airflow. After the airflow merges, the wind speed becomes approximately 21 m/s. This is because the air pressure in the area where the air flow rate is fast is small, and the external atmospheric pressure will affect these two air flows so that the air flows attract each other and shift in parallel directions. Although the direction of these two airflows is affected after approaching, after the air flow of the nozzle behind the jet direction passes through the nozzle in front, it is greatly affected by another air flow, and the direction change is more obvious. The air flow direction of the nozzle in front shifts less, and because of the influence of inertia, the direction of the two air flows will shift downward and flow along an arc after fusion.

Continuing to simulate the flow field of three nozzles, the influence of one air flow on the other two air flows is studied. Nozzles No. 1 and No. 2 are positioned and oriented in the same way as in the above simulation, and Nozzle No. 3 is added just below Nozzle No. 1 in the positive X-direction. The distances between two adjacent nozzles are equal. Because the spiral arrangement of nozzles in the simulated fog curtain device is gradually enlarged, the included angle of the nozzle injection direction is gradually reduced. The included angle between nozzle No. 3's injection direction and the X axis is -35˚, as shown in Fig 6. The airflow from nozzle No. 2 still shifts downward after approaching nozzle No. 1 and is affected by two airflows, and the airflow at this point shifts more obviously than the above simulation results. The airflow from nozzle No.1 is affected by the jet airflow from nozzle No.3, causing the airflow that should have been horizontally emitted there to shift downward. Although the airflow in this area is affected by airflow in two directions at the same time, it can be seen from the simulation results of two nozzles that the influence of the front-end airflow on the direction of the back-end airflow is greater than that of the back-end airflow on the front-end airflow, so the airflow from nozzle No.1 is affected by nozzle No. 3 and then inclines downward after passing through nozzle No. 3. Nozzle 3 is offset upward by the upper airflow, but the offset angle is minimal. After a certain distance, the air flow of the three nozzles will still merge into a whole, the wind speed of the three air flows after fusion is approximately 19 m/s, and the downward offset angle is larger than the air flow from 2 nozzles.

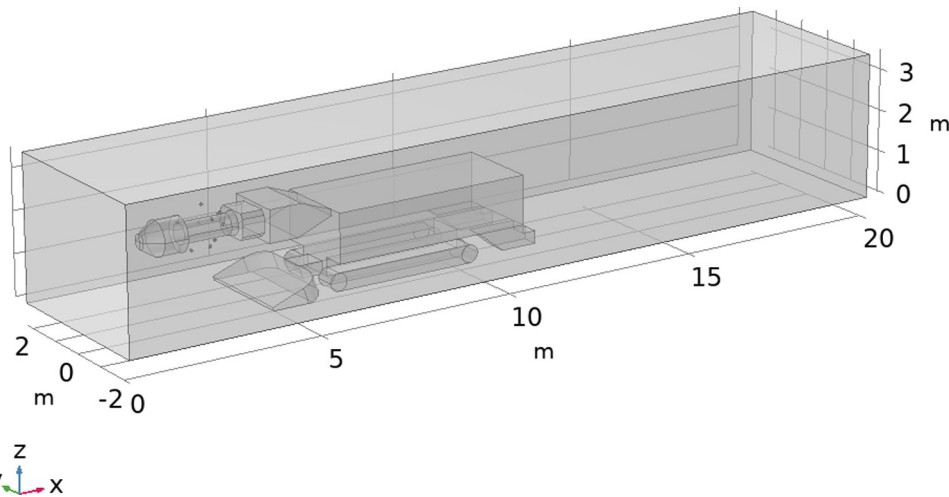

**Fig 7. Field application model diagram of the spraying device.**

## 4.2 Simulation of the airflow field and particle trajectory in field application

Taking the heading face of a mine as a prototype, the geometric model of the roadway with a rectangular cross-section that is 20 m, 4.5 m and 3.5 m in length, width and height is established at a 1:1 scale, and the roadway boring machine is established in the roadway. To simulate the spraying effect of the device in the field application, 10 pneumatic nozzles arranged according to two spiral layers with the cutting arm of the roadheader as the axis, and the spraying direction was tangent to the spiral trajectory. The physical model is established as shown in Fig 7.

Fig 8 shows the main view of the airflow streamline and flow direction. It can be seen from the figure that when the two-layer spiral fog curtain device is operated, a cyclone curtain

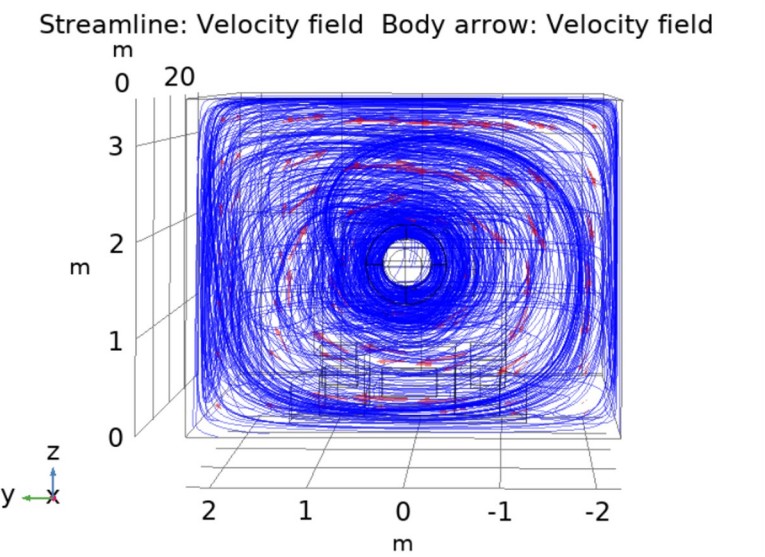

**Fig 8. Main view of the airflow streamline and direction.**

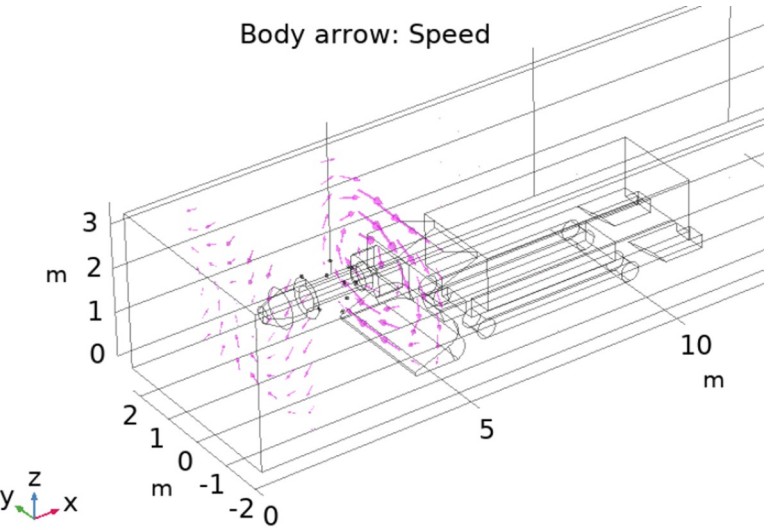

**Fig 9. Airflow trend chart.**

around the roadheader cutting arm will be formed in the roadway, and the cyclone covers the whole roadway section. It can be seen from the direction of the arrows that the cyclone wind direction is a spiral with the cutting arm as the axis. This is because the nozzles are distributed spirally around the cutting arm, and the jet directions of each nozzle are different and interact with each other so that the airflow in the tunneling roadway flows continuously in a spiral shape.

As seen from the direction of the airflow arrow in Fig 9, the two-layer spirally arranged nozzle structure makes the device work to form two layers of cyclone air curtain rotating around the cutting arm. Because of the low air pressure in the area with high wind speed, it is easier to gather water mist, which helps forma a two-layer mist curtain and realizes efficient dust control at the heading face.

The calculation results of the drag and turbulence models are coupled to calculate the trajectory of water mist particles, and the change in droplet size is calculated by the K-H breaking model. The number of droplets sprayed by each nozzle is 50 within 1 s. The atomization simulation results show the droplet distribution and movement trajectory and analyze the characteristics of the water mist distribution and its influence on dust removal. Fig 10 shows the

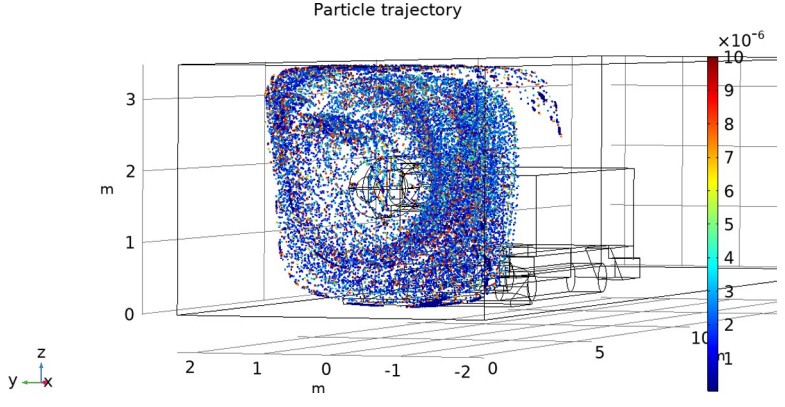

**Fig 10. Particle trajectory.**

water mist particle trajectory of the device while spraying. Under the influence of cyclones, a rotating mist curtain can cover the dust source. The shape of the mist curtain is roughly the same as that of the nozzle arrangement, and it has a two-layer spiral shape. This shows that the device can make the water mist rotate to form multiple layers to block the diffusion of dust. When the initial particle size of water mist is set to 10 μm, the water mist will be broken by the rotating airflow, and the particle size is mostly distributed over 1~4 μm. Smaller water mist is helpful to increase the contact opportunity with dust and to improve the sedimentation effect.

## 5. Experimental study of a multilayer spiral mist curtain device for dust removal

### 5.1 Overview of the experimental system

In this experiment, a test shed with a length of 4 m, a width of 3 m and a height of 3 m is used as a simulated tunneling roadway: the front area simulates the dust production state during the tunneling process of the roadheader, and the back area simulates the spraying process of the device. This experiment constructs a multilayer spiral fog curtain dust removal simulation device suitable for heading faces. The simulation device consists of a spiral pipeline support, water supply pipe, air supply pipe, water pump, air compressor, water tank and 10 pneumatic nozzles. The structure of the pipe support for fixing the nozzle, water supply pipe and air supply pipe is the same as that of the nozzle in the numerical simulation. A dust generating device is arranged at one end of the test shed and is composed of an air duct, a funnel and a blower connected at the back end. Pulverized coal is introduced into the air duct by a funnel, and the pulverized coal is introduced into the shed by starting the blower. The other end is a multilayer spiral fog curtain dust removal device simulated to be installed on the roadheader. The air compressor provides wind power to spray air from the nozzle, and together with the water pump, it provides the spraying power for the spiral fog curtain dust removal device. The spray device is 2.5 m away from the dust source and faces the dust source. The design of the multilayer spiral fog curtain test device is shown in Fig 11.

### 5.2 Layout of the experimental test equipment

As shown in the layout of a similar test platform in Fig 12, the main equipment used in the test is the following: 10 SK508 pneumatic nozzles; a water pump; an air compressor; a water supply pipe; an air supply pipe connecting the 10 nozzles; spiral pipe support; dust generating device; pulverized coal; and the dust concentration tester. When spray dust removal is adopted, field workers must stay away from the dust generation point, and dust easily gathers at the rear corner, so the measuring point is set as the leftmost rear corner of the test shed. A dust concentration tester is placed at the measuring point, and the measuring method is adjusted to perform

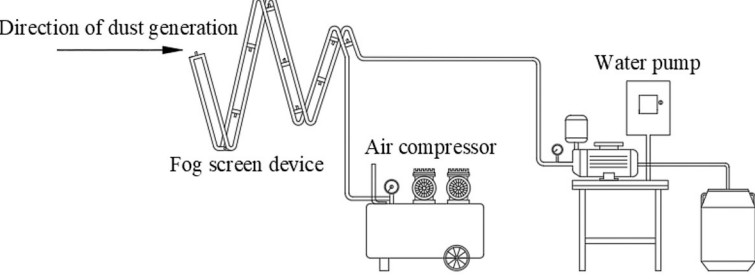

**Fig 11. Design of multilayer vortex fog curtain test model.**

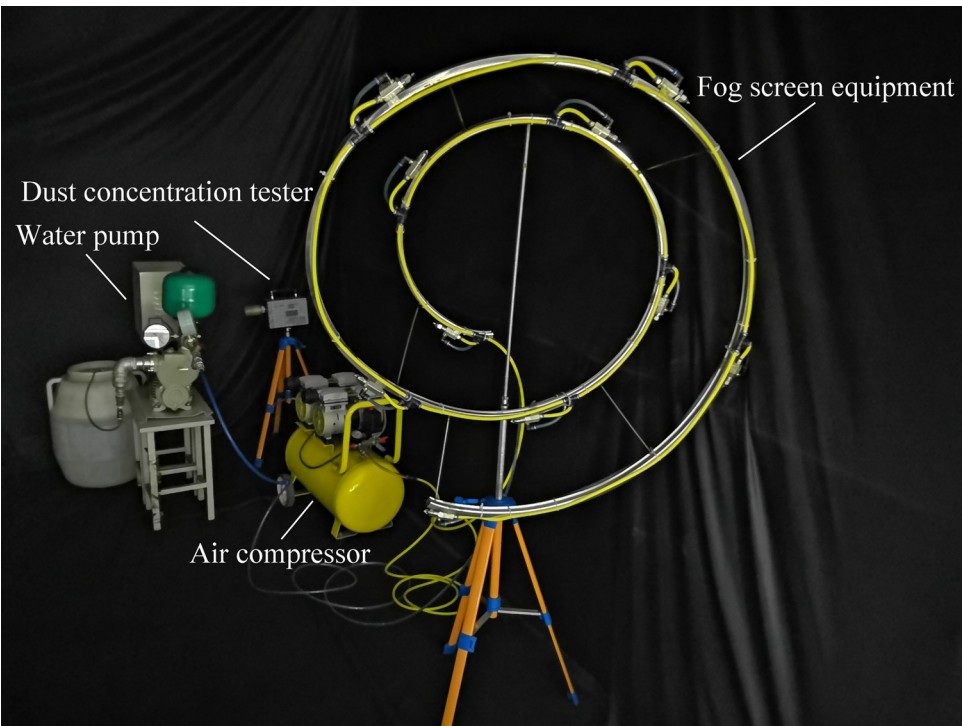

**Fig 12. Layout of a similar test platform.**

measurements every minute. In the spray process, the water pressure of the water pump is set to 0.12 MPa, and the air pressure of the air compressor is set to 0.52 MPa.

## 5.3 Experimental study of dust removal rates using multilayer spiral mist curtain devices

**5.3.1 Dust removal rate test procedure.** Three tests were carried out using 1.5 kg of pulverized coal per test. Each test used a different dust removal method, and the other conditions and parameters were the same. The dust removal methods used were natural sedimentation, traditional spray dust removal and multilayer spiral fog curtain dust removal. The traditional spray dust removal scheme used in this experiment arranges the nozzles in a ring, the angle is adjusted to the direction of radiating forward and outward from the center of the circle, and the spray angle is 60°. Approximately 45 s after the blower is started, the dust concentration change parameters $C_0$, $C_1$, $C_2$ and $C_3$ of each group are measured, where $C_0$ is the dust concentration measured by the dust concentration tester immediately after the dust emission, and $C_1$, $C_2$ and $C_3$ are the dust concentrations measured by the tester at 1 min, 2 min and 3 min after the dust emission. The dust concentration at each time period was recorded for each test group, and the dust removal rate of each test group was calculated when dust removal was carried out for 3 min. The formula is as follows:

$$v_n = \frac{C_0 - C_n}{t} \tag{17}$$

where $v_n$ is the dust removal rate in for stage n, $C_0$ is the initial dust concentration, and $C_n$ is the dust concentration at n minutes after completion of dust generation.

**Table 2.** Comparison of the dust concentration changes and dust removal rates at different times under three dust removal methods.

| dust removal mode | Concentration and rate | $C_0$ (mg/m³) | $C_1$ (mg/m³) | $C_2$ (mg/m³) | $C_3$ (mg/m³) | $\bar{v}$ Mg/(m³ s) |
|---|---|---|---|---|---|---|
| Natural dust sedimentation | Total dust | 469.54 | 378.97 | 307.06 | 248.84 | 1.226 |
| | Exhaled dust | 33.56 | 24.73 | 19.88 | 15.32 | 0.101 |
| Traditional spray dust removal | Total dust | 472.28 | 163.35 | 77.27 | 30.99 | 2.452 |
| | Exhaled dust | 34.15 | 16.89 | 7.86 | 2.79 | 0.174 |
| Multilayer spiral mist curtain for dust removal | Total dust | 471.36 | 45.93 | 24.81 | 3.42 | 2.600 |
| | Exhaled dust | 34.85 | 4.68 | 3.21 | 0.91 | 0.189 |

**5.3.2 Experimental results and analysis of dust removal rates.** The test results are summarized in Table 2, and the change charts of the total dust and exhaled dust concentrations are drawn, as shown in Figs 13 and 14.

From the data in Table 2, it can be seen that the initial total dust and exhaled dust concentrations of the three groups are close to 470 mg/m³ and 34 mg/m³ after 1.5 kg coal dust is released. After 1 min of natural dust reduction, the total dust concentration in the space is 378.97 mg/m³, and the exhaled dust concentration is 24.73 mg/m³. Among the three test methods, the natural sedimentation dust removal rate was the lowest, and the average total dust and exhaled dust removal rates within 3 min were only 1.226 mg/(m³·s) and 0.101 mg/(m³·s), respectively. After 1 min of traditional pneumatic spray dust removal, the total dust concentration decreased to 163.35 mg/m³, and the exhaled dust concentration decreased to 16.89 mg/m³. The average total dust and exhaled dust removal rates within 3 minutes were 2.452 mg/(m³·s) and 0.174 mg/(m³·s), respectively. However, the total dust and exhaled dust concentrations decreased to 45.93 mg/m³ and 4.68 mg/m³, respectively, after the multilayer spiral fog

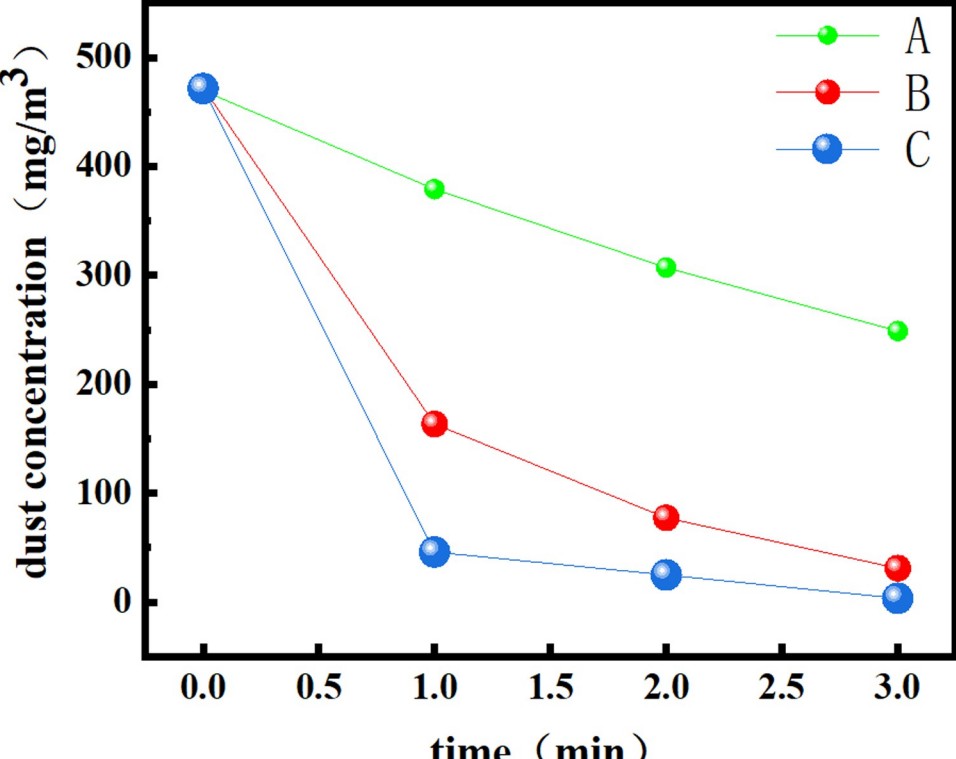

**Fig 13. Change in the total dust concentration measured for each of the three dust removal methods.**

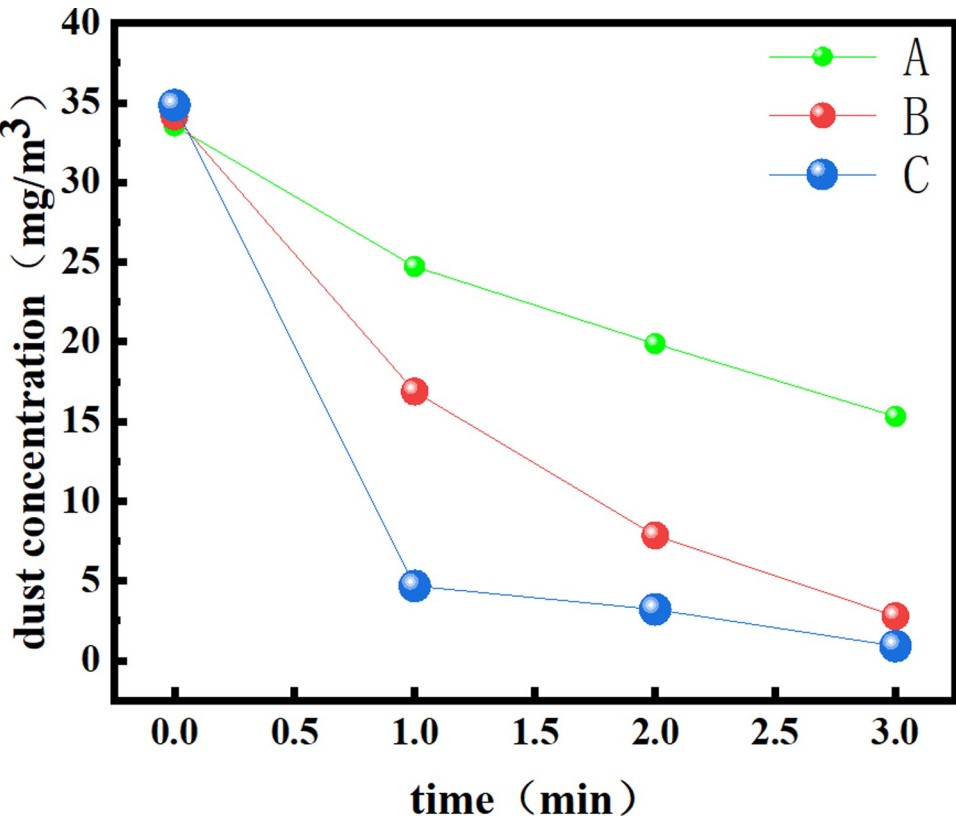

**Fig 14. Change in the exhaled dust concentration measured for each of the three dust removal methods.**

curtain dust removal device was turned on for 1 minute, and the average total dust and exhaled dust reduction rates reached 2.600 mg/(m³·s) and 0.189 mg/(m³·s), respectively, in 3 minutes. Compared with traditional spray, the dust removal ability of the multilayer spiral mist curtain is obviously better than that of traditional spray. After 3 minutes, the total dust concentration of the multilayer spiral mist curtain is reduced to 3.42 mg/m³, which is below the national standard of 4 mg/m³, and the exhaled dust concentration is also reduced to below 1 mg/m³.

A, B and C in Figs 13 and 14 are natural sedimentation, traditional spray dust removal and multilayer spiral fog curtain dust removal, respectively. It can be seen from the two figures that the concentration change of natural dust is relatively small for both total dust and exhaled dust, while the dust concentration for traditional spray and multilayer spiral fog curtain decreases rapidly in the first minute, drops to a relatively low concentration after one minute, and then tends to be low. This is because after the dust concentration decreases, the decrease in the number of particles in the unit space makes it more difficult to remove the dust. After the first minute of dust removal by using the multilayer spiral fog curtain, the decrease is the fastest, which shows that the multilayer spiral fog curtain device has the fastest dust removal rate and strong dust removal ability.

## 5.4 Experimental study of the dust removal efficiency of a multilayer spiral mist curtain device during continuous dust production

**5.4.1 Dust removal efficiency test procedure.**   These tests were carried out using 1.5 kg of pulverized coal in each of two groups: the traditional spray dust removal method and

**Table 3. Nomenclature table.**

| | |
|---|---|
| $v_n$ | Dust removal rate in for stage n |
| $C_0$ | Initial dust concentration, (mg/m³) |
| $C_1$ | Dust concentration at 1 minutes after completion of dust generation, (mg/m³) |
| $C_2$ | Dust concentration at 2 minutes after completion of dust generation, (mg/m³) |
| $C_3$ | Dust concentration at 3 minutes after completion of dust generation, (mg/m³) |
| $C_n$ | Dust concentration at n minutes after completion of dust generation, (mg/m³) |
| $\bar{v}$ | Average dust removal rate in 3 min, mg/(m³·s) |
| $\eta$ | Dust removal efficiency |
| $C_h$ | Dust concentration after 45 seconds of dust generation, (mg/m³) |
| A | Natural sedimentation |
| B | Traditional spray dust removal |
| C | Multilayer spiral fog curtain dust removal |

multilayer spiral fog curtain dust removal method in the dust emission process. The initial concentration $C_0$ of this test was the approximate average of the above test results, the total dust concentration was 470 mg/m³, and the exhaled dust concentration was 34 mg/m³. In both groups, the equipment was first started. After spraying stabilized, 1.5 kg of pulverized coal was loaded into the dust generating device to generate dust. After 45 seconds of dust generation, the dust concentration was measured immediately and recorded as $C_h$. According to the concentration data of the two groups of experiments, the dust removal efficiency $\eta$ of each group of experiments is calculated by formula (18), nomenclature and there descriptions are given on Table 3.

$$\eta = \frac{C_0 - C_h}{C_0} \tag{18}$$

where $\eta$ is the dust removal efficiency, $C_0$ is the initial dust concentration, and $C_h$ is the dust concentration measured after dust emission by two different dust removal methods.

**5.4.2 Experimental results and analysis.** The dust concentration test results are shown in Table 4.

According to the data in Table 3, when the traditional spray dust removal method is adopted, the concentrations of total dust and exhaled dust at the measuring points are controlled at 54.16 mg/m³ and 5.98 mg/m³, respectively, and the dust removal efficiency is 88.48% and 82.41%, respectively. However, when the multilayer spiral fog curtain is used for dust removal, the dust removal efficiency of the total dust at the measuring point is as high as 97%, the dust removal efficiency of the exhalation dust is close to 95%, and the concentrations are controlled at 14.06 mg/m³ and 1.93 mg/m³. It can be clearly seen that the dust removal efficiency of the multilayer spiral fog curtain is higher. Compared with the traditional spray dust removal efficiency, the dust removal efficiency of multilayer fog curtains is approximately 10%

**Table 4. Dust concentration and efficiency of two groups of tests after continuous dust emission.**

| Dust removal mode | Concentration and rate | Dust concentration $C_h$ (mg/m³) | Dust removal efficiency $\eta$ (%) |
|---|---|---|---|
| Conventional spray | Total dust | 54.16 | 88.48 |
| | Exhaled dust | 5.98 | 82.41 |
| Multilayer spiral fog curtain | Total dust | 14.06 | 97.01 |
| | Exhaled dust | 1.93 | 94.32 |

higher in terms of both the total dust and exhalation dust. This means that when using traditional spray dust removal, the airflow does not interact with each other to form a continuous airflow, which tends to cause disordered wind flow and slow diffusion of the water mist, which prevents it from colliding and combining with the dust more quickly, resulting in a lower dust removal rate. The multilayer spiral fog screen dust removal device uses spiral air flow to break the sprayed water mist more fully, further improves the atomization effect and the flow speed of the water mist, expands the effective dust isolation range, and increases the opportunity for fog droplets to capture dust. The spiral air flow forms a negative pressure zone where the dust is drawn into the spiral air flow due to the air pressure, giving the dust more opportunities to combine with the water mist. As a new dust removal method at the comprehensive excavation face, this device can form a multilayer radial fog curtain in the roadway to stop dust diffusion and has a more prominent dust removal and dust isolation effect than the traditional spray method. When applied to the heading face, the device can achieve good dust removal, prevent a large amount of dust from diffusing into the working area, and provide a good working environment for underground workers.

## 6. Conclusion

1. By simulating the airflow direction from a different number of nozzles in different positions, the simulation results show that when the nozzles are arranged in an arc and the injection direction is tangent to the arc, the jet airflow of each nozzle will interact with each other and make the airflow direction shift to the arc direction.

2. The coupling calculation of the turbulence model and fluid drag force model shows that the nozzles are arranged in a two-layer spiral pattern to achieve a multilayer spiral fog curtain, which can produce a two-layer fog curtain and prevent dust from diffusing to the workers' operation area. Under the action of a cyclone, the water mist in the fog curtain diffuses rapidly, forming a trumpet-shaped rotating water mist flowing around the central axis of the roadway, covering the dust source. Air flow can also play a certain role in breaking the water mist, which increases the collision probability between the water droplets and dust and improves the dust removal effect.

3. By comparing the dust removal rates of natural dust sedimentation, traditional spray dust removal and the multilayer vortex mist screen, the dust removal effect of multilayer spiral mist screen on the total dust and exhaled dust is obviously better than that of natural dust sedimentation and traditional spray dust removal, and the average total dust and exhaled dust removal rates within 3 minutes are 2.600 mg/(m$^3$·s) and 0.189 mg/(m$^3$·s), respectively. A total dust concentration of 470 mg/m$^3$ can be reduced below 4 mg/m$^3$, which is the national standard, and the exhaled dust concentration can be reduced below 1 mg/m$^3$ after only using a multilayer spiral fog curtain for 3 min. The correctness of the numerical simulation is also verified by similar experiments.

4. The dust removal efficiency of traditional spray and multilayer spiral fog curtains is obtained by continuous dust emission experiments, and their dust isolation effects are compared. Compared with a traditional pneumatic spray, the dust removal efficiency of a multilayer spiral fog curtain is approximately 10% higher for all dust and exhaled dust, and its dust removal and dust isolation effects are relatively good.

## Author Contributions

**Data curation:** Xiangxi Meng.

**Formal analysis:** Xiangxi Meng.

**Funding acquisition:** Deji Jing.

**Investigation:** Shaocheng Ge.

**Methodology:** Deji Jing.

**Project administration:** Deji Jing.

**Software:** Zhen Li, Tian Zhang, Xin Jia.

**Supervision:** Shaocheng Ge.

**Validation:** Zhen Li, Shaocheng Ge, Tian Zhang, Xiangxi Meng, Xin Jia.

**Visualization:** Tian Zhang, Xin Jia.

**Writing – original draft:** Zhen Li.

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
