## [Decision Letter · Decision Letter 0]

21 Feb 2022

PONE-D-22-02796Research on the mechanism of multilayer spiral fog screen dust removal at the comprehensive excavation facePLOS ONE

Dear Dr. Jing,

Thank you for submitting your manuscript to PLOS ONE. After careful consideration, we feel that it has merit but does not fully meet PLOS ONE’s publication criteria as it currently stands. Therefore, we invite you to submit a revised version of the manuscript that addresses the points raised during the review process.

We look forward to receiving your revised manuscript.

Kind regards,

Mohammad Mehdi Rashidi

Academic Editor

PLOS ONE

Journal Requirements:

"National Natural Science Foundation of China Youth Fund Grant (51704146); LiaoningProvincial Natural Science Foundation Project (2020-MS-304); Liaoning Provincial EducationDepartment Scientific Research Funding Project (LJKZ03). The authors gratefully appreciate andacknowledge the financial support."

"National Natural Science Foundation of China Youth Fund Grant (51704146); Liaoning Provincial Natural Science Foundation Project (2020-MS-304); Liaoning Provincial Education Department Scientific Research Funding Project (LJKZ03).The funders had no role in study design, data collection and analysis, decision to publish, or preparation of the manuscript."

Reviewers' comments:

Reviewer's Responses to Questions

**Comments to the Author**

1. Is the manuscript technically sound, and do the data support the conclusions?

Reviewer #1: Yes

Reviewer #2: Yes

2. Has the statistical analysis been performed appropriately and rigorously? 

Reviewer #1: Yes

Reviewer #2: Yes

3. Have the authors made all data underlying the findings in their manuscript fully available?

Reviewer #1: Yes

Reviewer #2: Yes

4. Is the manuscript presented in an intelligible fashion and written in standard English?

Reviewer #1: Yes

Reviewer #2: Yes

5. Review Comments to the Author

Reviewer #1: In this paper, a multilayer spiral fog screen dust removal method is proposed for the first time. This dust removal method creates a multilayer fogging dust removal screen and increases the chance of contact between the fog droplets and the dust. It is innovative. The research discusses the movement law of wind flow under interaction, conducts a numerical simulation of multi-physical field coupling for the application of multilayer spiral fog screen device, and establishes a similar experimental platform to study and test the dust control performance of the multilayer spiral fog screen dust control method, and achieves good results, which has guiding significance for dust control at the comprehensive excavation face. This paper seems to solve a problem that is in production practice. I have a small suggestion though. The paper should discuss the relevant research in more detail to better understand the novelty of the paper and the contribution to the research institution. The paper is considered for acceptance, with minor revisions recommended.

Modification Comments：

1. Some figures are not clear, e.g. Figure 12

2、Adjust the size of the images appropriately and also try to keep the size of all the images in the manuscript as uniform as possible

3、The size of the formulae in this manuscript should be consistent

4. The font size of the title of this manuscript should be adjusted to the font size specified in this journal

5. Please briefly explain the significance of the simulation of the phenomenon of simultaneous injection of different numbers of nozzles in this manuscript for this research work

6. The experimental summary of this manuscript shows that the multi-layer spiral mist curtain device has better performance than the traditional spraying, and its advantages should be added to the specific description

Reviewer #2: This manuscript proposes a dust removal method using a multilayer spiral fog curtain with potential application in the dust-producing areas in mines. Considering the dangers of high-concentration dusts for miners, researches on this field are very essential. The manuscript includes experimental and theoretical sections. In the experimental section, an equipment has been made and dust removal rate and efficiency have been measured. In the theoretical section, numerical simulation of spray airflow and particle trajectory has been performed based on fluid mechanics relations. The manuscript is good in general. I recommend acceptance with minor comments as follows:

1. The manuscript file doesn’t have page numbers and line numbers. It is hard to refer to a particular word or sentence when you cannot refer to line numbers.

2. Has the device been tested in a real mine environment?

3. In the present manuscript, numerical simulations investigate the fluid mechanics of the system and experimental section studies the efficiency of dust removal (mass transfer). For future studies, I recommend the authors to couple mass transfer relations with the fluid mechanics equations and predict the dust removal efficiency by numerical simulations.

4. In the introduction, authors should explain more about the innovation of their work.

5. It is recommended to add a Nomenclature table to the manuscript.

6. The quality and sharpness of some figures are not very good (for example: Figs. 10, 5 and 6).

7. In section 3.2, some equations don’t have equation numbers.

8. In the text, reference number should be inserted after the “author’s name et al.”.

9. In the text, please use the same format to refer to the figures (“Figure” or “Fig. “).

10. Please use a space before the first bracket of all in-text citations.

11. The font of some parts of the manuscript is different from the rest of the text (such as references, some tables, the first paragraph of the conclusion, equation numbers and etc.). Please correct this problem in the manuscript.

12. Some parts of the text aren’t justified (for example: references, the first paragraph of the section 3).

13. After all the headline numbers, the punctuation symbol is required. Some headlines also start with a lowercase letter that need to be corrected.

6. PLOS authors have the option to publish the peer review history of their article (what does this mean?). If published, this will include your full peer review and any attached files.

Reviewer #1: No

Reviewer #2: No

---

## [Author Response · Author response to Decision Letter 0]

16 Mar 2022

Dear Editors and Reviewers:

Thank you for your letter and for the reviewers’ comments concerning our manuscript entitled “Research on the mechanism of multilayer spiral fog screen dust removal at the comprehensive excavation face”. Those comments are all valuable and very helpful for revising and improving our pa per, as well as the important guiding significance to our researches. We have studied comments carefully and have made correction which we hope meet with approval. The paper is revised in word's revision mode, and the red area indicate the revised portion. The main corrections in the paper and the responds to the reviewer’s comments are as flowing:

Response to Editor

1.The manuscript has complied with PLOS ONE style requirements, including file naming requirements.

2. Any text relating to funding has been removed from the manuscript.

3. Corresponding author has ORCID iD and validates it in Editorial Manager.

4. The reference list has been checked to ensure it is complete and correct.

Response to Reviewer#1

1. Fig 12 has been replaced with a clear picture in this manuscript

2. The images has been resized appropriately in the manuscript.

3. The size of the formulae in this manuscript has been appropriately resized.

4. The font size of the title of the manuscript has been adjusted to the font size specified in this journal.

5. By simulating the airflow direction from a different number of nozzles in different positions, the simulation results show that when the nozzles are arranged in an arc and the injection direction is tangent to the arc, the jet airflow of each nozzle will interact with each other and make the airflow direction shift to the arc direction.The preliminary demonstration of an important reason for the spiral fog screen produced by the device provides a theoretical basis for the subsequent spray simulation and testing of a multilayer spiral fog screen device.

6. The multilayer spiral fog screen dust removal device uses spiral air flow to break the sprayed water mist more fully, further improves the atomization effect and the flow speed of the water mist, expands the effective dust isolation range. The spiral air flow forms a negative pressure zone where the dust is drawn into the spiral air flow, increasing the chances of the droplets catching the dust. This device can form a multilayer spiral fog curtain in the roadway to stop dust diffusion and has a more prominent dust removal and dust isolation effect than the traditional spray method. The advantages of these multilayer spiral mist screen devices over conventional spray devices have been added to the section 5.4.2 of the manuscript.

Response to Reviewer#2

1. Page numbers and line numbers have been added to the manuscript.

2. The device has been tested in a real mine environment.

3. Thank you for your valuable suggestions and we will focus on the prediction of dust removal efficiency through numerical simulations in future studies.

4. An explanation of the innovation of the scholars' work has been added to the introduction.

5. A nomenclature table has been added to the section 5.4.1 of the manuscript, with additions and corrections to the explanation of variables in the formulae.

6. Fig 10、5 and 6 has been replaced with a clear picture in this manuscript 

7. Equation numbers have been added to some of the equations in the section 3.2.

8. Reference numbers have been inserted in the text after " author’s name et al.".

9. The same format ("Fig.") has been used in the text to refer to the figures.

10. A space has been used before the first bracket of all in-text citations.

11. The font has been corrected for certain parts of the manuscript where the font differs from the rest of the text (such as references, some tables, the first paragraph of the conclusion, equation numbers and etc.).

12. The rationale of the references cited has been added to the first paragraph of the section 3 of the manuscript, where the references cited are relevant to the content of the study.

13. After all the headline numbers, the punctuation symbol has been added，and some headings beginning with lowercase letters have been corrected.

---

## [Decision Letter · Decision Letter 1]

25 Mar 2022

Research on the mechanism of multilayer spiral fog screen dust removal at the comprehensive excavation face

PONE-D-22-02796R1

Dear Dr. Jing,

We’re pleased to inform you that your manuscript has been judged scientifically suitable for publication and will be formally accepted for publication once it meets all outstanding technical requirements.

Kind regards,

Mohammad Mehdi Rashidi

Academic Editor

PLOS ONE

Additional Editor Comments (optional):

Reviewers' comments:

Reviewer's Responses to Questions

**Comments to the Author**

1. If the authors have adequately addressed your comments raised in a previous round of review and you feel that this manuscript is now acceptable for publication, you may indicate that here to bypass the “Comments to the Author” section, enter your conflict of interest statement in the “Confidential to Editor” section, and submit your "Accept" recommendation.

Reviewer #1: All comments have been addressed

Reviewer #2: All comments have been addressed

2. Is the manuscript technically sound, and do the data support the conclusions?

Reviewer #1: Yes

Reviewer #2: Yes

3. Has the statistical analysis been performed appropriately and rigorously? 

Reviewer #1: Yes

Reviewer #2: Yes

4. Have the authors made all data underlying the findings in their manuscript fully available?

Reviewer #1: Yes

Reviewer #2: Yes

5. Is the manuscript presented in an intelligible fashion and written in standard English?

Reviewer #1: Yes

Reviewer #2: Yes

6. Review Comments to the Author

Reviewer #1: The authors have addressed all my comments, therefore I suggest to accept the paper in the present form.

Reviewer #2: Thank you for the modifications to the manuscript. I recommend acceptance but please correct the following comments in the final version of manuscript.

1. For in-text citations, it isn’t required to mention the author’s full name. Please just use the “author’s family name et al.”.

2. In the revised manuscript (page 16, line 420), please use the lowercase letters for the word of “Wind”.

7. PLOS authors have the option to publish the peer review history of their article (what does this mean?). If published, this will include your full peer review and any attached files.

Reviewer #1: No

Reviewer #2: No

---

## [Editor Report · Acceptance letter]

31 Mar 2022

PONE-D-22-02796R1 

Research on the mechanism of multilayer spiral fog screen dust removal at the comprehensive excavation face 

Dear Dr. Jing:

I'm pleased to inform you that your manuscript has been deemed suitable for publication in PLOS ONE. Congratulations! Your manuscript is now with our production department. 

Kind regards, 

on behalf of

Professor Mohammad Mehdi Rashidi 

Academic Editor

PLOS ONE